# A Pillar-Based High-Throughput Myogenic Differentiation Assay to Assess Drug Safety

**DOI:** 10.3390/molecules26195805

**Published:** 2021-09-25

**Authors:** Kyeong Hwan Ahn, Sooil Kim, Mihi Yang, Dong Woo Lee

**Affiliations:** 1Department of Biomedical Engineering, Konyang University, Daejeon 35365, Korea; 20806502@konyang.ac.kr (K.H.A.); sooil234@dgist.ac.kr (S.K.); 2Department of Toxicology, College of Pharmacy, Sookmyung Women’s University, Seoul 04310, Korea; 3Central R & D Center, Medical & Bio Decision (MBD) Co., Ltd., Suwon 16229, Korea

**Keywords:** pillar strip, myogenic differentiation, developmental toxicity, C2C12, Matrigel coating plate

## Abstract

High-throughput, pillar-strip-based assays have been proposed as a drug-safety screening tool for developmental toxicity. In the assay described here, muscle cell culture and differentiation were allowed to occur at the end of a pillar strip (eight pillars) compatible with commercially available 96-well plates. Previous approaches to characterize cellular differentiation with immunostaining required a burdensome number of washing steps; these multiple washes also resulted in a high proportion of cellular loss resulting in poor yield. To overcome these limitations, the approach described here utilizes cell growth by easily moving the pillars for washing and immunostaining without significant loss of cells. Thus, the present pillar-strip approach is deemed suitable for monitoring high-throughput myogenic differentiation. Using this experimental high-throughput approach, eight drugs (including two well-known myogenic inhibitory drugs) were tested at six doses in triplicate, which allows for the generation of dose–response curves of nuclei and myotubes in a 96-well platform. As a result of comparing these F-actin (an actin-cytoskeleton protein), nucleus, and myotube data, two proposed differentiation indices—curve-area-based differentiation index (CA-DI) and maximum-point-based differentiation index (MP-DI) were generated. Both indices successfully allowed for screening of high-myogenic inhibitory drugs, and the maximum-point-based differentiation index (MP-DI) experimentally demonstrated sensitivity for quantifying drugs that inhibited myogenic differentiation.

## 1. Introduction

Animal or mammalian cells grown in culture have been used to assess in vitro or in vitro toxicity, respectively. Although animal studies provide more precise physiological conditions for the evaluation of toxic chemicals, animal studies have some restrictions (e.g., cost, ethical issues, extrapolation) [1]. To avoid these restrictions, many research groups have developed alternatives to animal studies using human cells or tissues grown in culture [2]. Particularly, the cell-differentiation assay is a valuable technology that can be used to predict in vivo developmental toxicity in stem cells [3,4,5,6,7,8]. Since the C2C12 myoblast cell line was developed in 1977, it has been commonly used in myogenic differentiation assays [8,9,10,11,12,13]. Skeletal muscles account for approximately 40% of body mass and are essential for both locomotion and whole-body metabolism [14,15]. Loss of skeletal muscle mass during aging or pathogenesis is a well-known contributor to morbidity and mortality. Thus, an ideal myogenic differentiation assay is needed to screen and characterize myogenic inhibition of new drugs, which can be potentially toxic compounds. Conventionally, the myogenic differentiation assay is conducted in 96-well or 12-well plates with the growth and differentiation of C2C12 cells. To identify myogenic differentiation with a microscope, the myosin of C2C12 cells needs to be immunostained with markers, such as MF20 (Myosin 4 Monoclonal Antibody, 1:1000, #14-6503-80, eBioscience™, Invitrogen, Carlsbad, CA, USA) [16,17,18,19]. However, immunostaining includes multiple steps (e.g., fixation, blocking, washing). Thus, the experimental yield with those methods is limited because (i) they are tedious with many steps, (ii) pipetting during solution replacement may cause cellular damage, and (iii) multiple washes result in the loss of a high proportion of cells.

To overcome these limitations, a pillar-strip approach was proposed to speed up the process of solution replacement. C2C12 cells from an immortalized mouse myoblast cell line were attached to the pillar strips and the pillar strips moved from well to well for immunostaining. Thus, the pillar strip can be easily washed without cell damage by moving the pillar strip at once. The pillar strip was coated with Matrigel to improve the differentiation of C2C12 [20]. Our pillar strip has eight pillars with a diameter of 2 mm, as shown in Figure 1. C2C12 cells adhered and differentiated on the Matrigel-coated surface. The pillar strip containing C2C12 was easily moved to new growth media, differentiation media, and immunostaining reagents in a high-throughput manner, as shown in Figure 2. The high-throughput manner could generate cell growth (F-actin and nucleus) and cell differentiation (myotube) data according to the drug concentration of multiple drugs at once. Pillar strips compatible with 96-well plates conduct myogenic differentiation assay with six different doses and three replicates of each dose, which yield dose-response curves (DRC) for F-actin, nuclei, and myotubes. As a result of comparing these nuclear and myotube data, two new differentiation indices were proposed to quantitatively analyze drug inhibition of myocyte differentiation. According to the comparison method of nuclei and myotubes, the curve-area-based differentiation index (CA-DI) and maximum-point-based differentiation index (MP-DI) were calculated. To verify the potential of this proposed pillar-strip approach, relative inhibition of C2C12 differentiation was quantified for eight drugs previously known to inhibit muscle cell differentiation or cell proliferation.

## 2. Results

### 2.1. Myotube Expression

To improve C2C12 differentiation, pillar surfaces were coated with Matrigel. Figure 3a reveals myotube imaging on non-coated and Matrigel-coated pillarstrips. Under the same conditions, more C2C12 cells on the Matrigel-coated surfaces were differentiated compared to those on the non-coated surface. Red fluorescence is specific for F-actin (a protein of the actin cytoskeleton). Green fluorescence is a marker of myotube (myosin); blue fluorescence marks the presence of nuclei. Green fluorescence areas were extracted and calculated as shown in Figure 3b. Green areas were normalized with those of the Matrigel-coating condition at day 10. The myotube areas were 5-fold higher in Matrigel-coating pillar strips compared with non-coated strips. In Western blots (Figure 3c), myosin 4 expression is two-fold higher in Matrigel-coating conditions compared with non-coated strips. Thus, we optimized C2C12 differentiation conditions in the proposed pillar-strip assay to quantify myogenic differentiation.

Before starting the experiment using pillar strips, we tried to coat wells of 96 well plates with Matrigel and conducted the C2C12 differentiation experiment. However, most of the cells are clustered at the edges of the wells, and C2C12 is easily detached, especially during immunostaining at multiple aspiration and filling steps. This detachment may vary depending on the user’s skill and equipment.

### 2.2. Myogenic Differentiation Assay

Eight drugs were tested in triplicate at six doses using the pillar strip compatible with 96-well plates (Figure 1b). Figure 4 shows the dose–response curves for F-actin, nuclei, and myotube staining according to the drug concentration of eight drugs. F-actin staining show whole-cell skeleton-related cell proliferation and differentiation. Nuclei staining detects proliferating cells and is an indirect measure of efficacy for each of the drugs tested. A drug with higher efficacy would inhibit cell growth leading to fewer nuclei-stained cells. However, inhibition of cellular differentiation can occur in highly viable cells or proliferating cells before inhibiting cell proliferation. Thus, a comparison of microtubes and nuclei is important to quantify cell differentiation. According to the comparison method of nuclei and myotubes, two differentiation indices, the curve-area-based differentiation index (CA-DI) and maximum-point-based differentiation index (MP-DI), were calculated. 

CA-DI was calculated as the ratio of the gap area between nuclei and myotubes per nucleus area-under-the curve as shown in Equation (1). As shown in Figure 4d, Doxorubicin shows a similar area under the curve (AUC) of nuclei and myotubes. Doxorubicin showed high cytotoxicity and did not inhibit myogenic differentiation at very low doses. Therefore, the gap between curves of nuclei and myotubes is small, and CA-DI is also low. In the case of rapamycin in Figure 4g, myotube expression was dramatically reduced at low doses. The gap area between nuclei and myotubes is very large, and CA-DI is also high. Thus, rapamycin, a well-known myogenic inhibitory drug [21], demonstrates inhibition of myogenic differentiation in the proposed pillar-based assay. In the case of Regorafenib, Lenvatinib, and Cabozantinib, the gap area between nuclei and myotubes is quite small, and CA-DI is low (from 20% to 40%). However, those drugs showed very low expression of myotube with high nuclei expression at specific concentrations. These events occurred at high doses with Regorafenib (Figure 4b) and low doses with Lenvatinib (Figure 4c) and Carbozantinib (Figure 4e). We found this issue because many data were collected at different drug doses using pillar-based high-throughput screening to perform immunostaining in a high-throughput manner. 

Therefore, we developed a novel differentiation index (maximum-point-based differentiation index, MP-DI) based on high-throughput screening that takes into account the maximum point of myogenesis inhibition at specific drug concentrations (see Equation (2) and Figure 5a). 

The maximum-point-based differentiation index (MP-DI) was calculated as the ratio of the maximum distance point among many drug doses per half diagonal length (50√2) of the relative nuclei area versus myotube area graphs as shown in Figure 5a. The comparison graphs of relative nucleus and myotube areas are generated. Relative nuclei and myotube areas represent cell proliferation and differentiation, respectively. Identically, relative nuclei and myotube areas were similar at various different drug concentrations (i.e., mean dashed line in Figure 5). Dots under the dashed line means specific concentration drugs inhibited myotubes more than nuclei; the distance from the dashed line indicates the degree of cell differentiation inhibition. For sorafenib (Figure 5a), 10 nM maximized cell differentiation while other doses revealed similar rates of inhibition for proliferation and differentiation. This suggests that sorafenib mainly inhibits cell proliferation. In Figure 5g, rapamycin and ethanol, well-known inhibitory drugs, all points (all different doses) are under the dashed line, suggesting that rapamycin and ethanol mainly inhibit cell differentiation. The pillar-based high-throughput screening could be leveraged to conduct many tests, including many different doses of drugs, to quantify myogenic differentiation inhibition. Among many points (dose of drug), the maximum distance from the dashed line could represent differentiation inhibition. Thus, the maximum-point-based differentiation index (MP-DI) was calculated by dividing the maximum distance with half diagonal length (50√2) of the relative nuclei area versus myotube area graph. Sorafenib and doxorubicin show short maximum distances, while rapamycin and ethanol (well-known myogenic inhibitory drugs) show long maximum distances. Thus, the MP-DI is low (20~30) in sorafenib- and doxorubicin-tested samples, while rapamycin and ethanol show high MP-DI (50~60). In the case of Regorafenib, Lenvatinib, and Cabozantinib, MP-DI values are high (20~50), which is similar to ethanol and rapamycin. It causes MP-DI considering only the maximum differentiation inhibition point. For example, 10 µM Regorafenib inhibited 100% myotube while the nucleus area is 38% in Figure 4b and Figure 5b. At lower than 10 µM Regorafenib, the relative myotube area and nucleus area are similar. Therefore, CA-DI is 26.52 and MP-DI is 42.3, indicating that MP-DI can more sensitively measure inhibition of muscle differentiation. Figure 6 shows the comparison of CA-DI and MP-DI; MP-DI reveals more sensitive results for Regorafenib, Lenvatinib and Cabozantinib. In Table 1, the MP-DI of rapamycin is revealed to be 60.2%, significantly higher when compared with sorafenib (28.4%). Previous studies demonstrated that rapamycin [21] inhibits cellular differentiation by blocking mTOR activity, while sorafenib [22] inhibits ATP activity leading to apoptosis. Doxorubicin has been shown to prevent the formation of DNA double helices during cell proliferation, thus inducing apoptosis [23], also had a very low MP-DI. In previous studies, regorafenib [24] and Lenvatinib [25], Cabozantinib [26], and 5-FU [27] reported differentiation inhibition, which is correlated with MP-DI. Thus, the proposed pillar-strip approach to stain nuclei and myotubes was capable of quantifying the inhibition of cell differentiation using the MP-DI as well as CA-DI.

## 3. Discussion

High-throughput, pillar-strip-based assays have been proposed to screen for drug safety against developmental toxicity. Muscle cell culture and differentiation were allowed to occur at the end of a pillar strip (eight pillars) compatible with commercially available 96-well plates. This pillar strip efficiently conducted immunostaining without significant loss of cells by allowing for easy movement of the pillar strip containing cells. Using these experiments, we proposed a differentiation toxicity index based on the abundance of nuclei and microtubes using a high-throughput approach. Ethanol and rapamycin, with well-known differentiation toxicity, demonstrated high differentiation toxicity indices in this pillar-strip-based approach. On the other hand, sorafenib, Regorafenib, Doxorubicin, and 5-FU showed differentiation toxicity, but at the same time, cytotoxicity was also shown; therefore, they showed low differentiation indices. Thus, the present pillar-strip-based approach is deemed suitable for monitoring high-throughput myogenic differentiation.

## 4. Materials and Methods

### 4.1. Cell Culture

The C2C12 cell line was purchased from the Korean Cell Line Bank (Seoul, South Korea). A549 cells, adenocarcinomic human alveolar basal epithelial cells, were also cultured in DMEM medium (Gibco) supplemented with 10% fetal bovine serum (FBS). The two cell lines were maintained at 37 °C in a 5% CO_2_-humidified atmosphere and passaged every four days. Normally, the C2C12 cell line was used in fewer than 20 passages after being thawed from the frozen cell stocks.

### 4.2. Experiential Procedure

Pillar strips made of polystyrene were treated for 1 min at 80 mW power with plasma (Femto Science Plasma system cute, 80 W, 1 min). Plasma activates the polystyrene of pillar to enhance the adhesion of Matrigel. Matrigel (Corning, Tewksbury, MA, USA) was diluted 10 times; 1 µL was dispensed onto the surface of the pillars. The pillars were dried for 30 min on a clean bench. Cell mixtures were prepared (2 × 10^6^ cells/mL) and dispensed onto the pillars using a microarrayer (ASFA™ Spotter ST, Medical & Bio Device, Suwon, Korea) as shown in Figure 2a. Approximately 2000 cells were dispensed to each pillar of the pillar strip. To allow for cell attachment to pillars, cells were stabilized for 1.5 h in an incubator (Figure 2b). During this time, the plates were covered to prevent drying. The pillar strips with attached cells were moved to wells filled with growth media consisting of 10% fetal bovine serum in Dulbecco’s Modified Eagle’s Medium (DMEM) for 2 days to yield a cell density of 80% (Figure 2c). To reduce assay time, increasing seeding density up to 4000 cells/pillar may reduce C2C12 confluency (when the pillar surface is more than 80% filled with C2C12) time. Media were changed from growth media to differentiation media to induce differentiation and cultured for 5 days in the presence of four experimental drugs in an incubator (Figure 2d). The cells exposed to the drug were stained (Figure 2f). Stained cells were scanned using an optical microscope (Nikon, Minato City, Tokyo, Japan) as shown in Figure 2g to quantify the abundance of myotubes and nuclei. Scanned images (Figure 2h) were evaluated with image-analysis software (ASFA™ Ez SW, Medical & Bio Device).

### 4.3. Immunofluorescence Staining

As shown in Figure 2f, C2C12 cells differentiated on the pillar strip were fixed using a 4% paraformaldehyde solution (PFA, Biosesang, Korea) for 1 h. After fixation, pillar strips were moved into a permeabilizing and blocking solution (5% bovine serum albumin, BSA) in phosphate-buffered saline (PBS) containing 0.3% Triton-X) for 3 h. Each micropillar was incubated overnight at 4 °C with the first antibody staining solution. The first antibody staining solution was prepared by adding the first antibody (MF20, Myosin 4 Monoclonal Antibody, 1:1000, #14-6503-80, Invitrogen, Carlsbad, CA, USA) into the blocking solution (5% Bovine Serum Albumin, BSA) at a ratio of 200:1. Stained pillar strips were washed with a blocking solution for 20 min. The second antibody staining solution was prepared by adding secondary antibody labeled with green fluorescence (A32723, Invitrogen), DAPI (Hoechst 33342, Invitrogen), and F-actin (F-actin phalloidin, Thermofisher Scientific) into the blocking solution at a ratio of 1000:1 for both. Stained pillar strips were washed using blocking solution for 20 min.

### 4.4. Image Processing and Data Analysis

Nuclei, myotubes, and F-actin of C2C12 on the Matrigel-coated pillars were stained with different color fluorescence (Blue, Green, and Red). Red fluorescent was used for filamentous actin (F-actin), which is composed of a cell cytoskeleton, to characterize C2C12 cell morphology. Green fluorescent dye was used for myotubes (myosin) in C2C12, which indicated C2C12 differentiation. Blue fluorescent dye was used for cell nuclei. An automatic optical fluorescence scanner (ASFA™ Scanner ST, Medical & Bio Device, Suwon, South Korea) was used to measure the red, green, and blue fluorescence intensities using an 8-bit code among the RGB codes (0–255). Using automatic area identification in ASFA™ Scanner ST, the abundance of myotubes was calculated using the size of the green area. Cell proliferation was quantified using the size of the blue area (Figure 3). The relative nuclei (Blue) and myotubes (Green) values were normalized to their corresponding controls (no drug treatment). Dose–response curves were obtained by plotting the expression values according to the dose of the drugs in GraphPad Prism 5, as shown in Figure 4. The area under the curve (AUC) values were calculated automatically in the XY analysis completed with the GraphPad Prism software. Loss of myotube formation can be affected by both inhibitions of differentiation and cell death due to inhibition of proliferation. Therefore, the AUC (green area) for myotube formation does not directly indicate cell differentiation. Comparison of microtube and nuclei is important to quantify cellular differentiation. According to the comparison method of nuclei and myotubes, two differentiation indexes, curve-area-based differentiation index (CA-DI) and maximum-point-based differentiation index (MP-DI), were calculated.

The curve-area-based differentiation index (CA-DI) was calculated as the ratio of the gap area between nuclei and myotubes (see Figure 4) per nuclei area-under-the-curve calculation as below.
(1)$$\mathrm{CA}\text{-}\mathrm{DI}\;[\%]=100-\frac{\mathrm{AUC}\;\mathrm{of}\;\mathrm{Myotube}}{\mathrm{AUC}\;\mathrm{of}\;\mathrm{Nucleus}}\times 100$$

A low differentiation index indicates that the drug dose does not inhibit cell differentiation even though the AUC for myotube formation is low.

The maximum-point-based differentiation index (MP-DI) was calculated as the ratio of the maximum distance point among many drugs dose per half diagonal length (50√2) of the nucleus area versus myotube area graph as shown in Figure 5. The MP-DI was calculated using the distance between the reference line (see block dashed line in Figure 5a) and the point using the formula below.
(2)$$\mathrm{MP}\text{-}\mathrm{DI}\;[\%]=\frac{\left|x-y\right|}{\sqrt{1^{2}+1^{2}}}\times \frac{1}{50\sqrt{2}}\times 100=\left|x-y\right|$$

x and y are the normalized fluorescence areas of the nucleus and myotube, and the reference line is defined as x − y = 0. (See Figure 5a). The longer the maximum distance, the better the inhibition of differentiation.

### 4.5. Western Blot Assay

Total C2C12 cell lysates were prepared using a cOmplete™ Lysis-M buffer solution (Roche Life Science, Mannheim, Germany). Protein extracts were resolved using 4–20% Mini-PROTEAN TGX™ Precast Protein Gels (Bio-Rad, Hercules, CA, USA) and transferred into iBlot^®^ PVDF gel Transfer Stack membranes (Thermofisher Scientific Korea Ltd., Seoul, Korea). After blocking non-specific binding sites for 1 h with 5% BSA in Tris-buffered saline containing 0.1% Tween-20 (TBS-T), membranes were incubated overnight at 4 °C with specific primary antibodies (Myosin 4 Monoclonal Antibody (1:1000, #14-6503-80, Invitrogen) and anti-beta actin (1:2000, Abcam, Cambridge, United Kingdom)) in accordance with the manufacturers’ instructions.

### 4.6. Statistical Analysis

All expression data for F-actin, myotubes, and nucleus were calculated as mean and standard deviation in triplicate replicates. To identify significant differences between myotube and nuclear expression, *p*-values are calculated. In Figure 5, *p*-values less than 0.01 *p*-value are marked with “*”, which are significantly different. *p*-values were calculated by Student t-test using GraphPad Prism 9.0. 

## Figures and Tables

**Figure 1 molecules-26-05805-f001:**
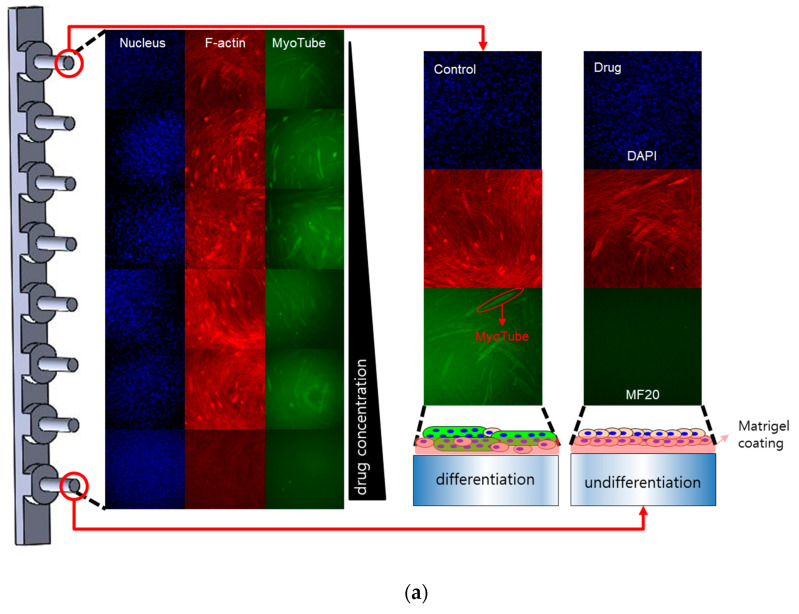
Schematic view of the pillar-based high-throughput myogenic differentiation assay. (**a**) A pillar strip was used to study inhibition of myogenic differentiation and (**b**) drug layout for the myogenic differentiation assay. Side wells were filled with PBS to prevent evaporating media during cell culture.

**Figure 2 molecules-26-05805-f002:**
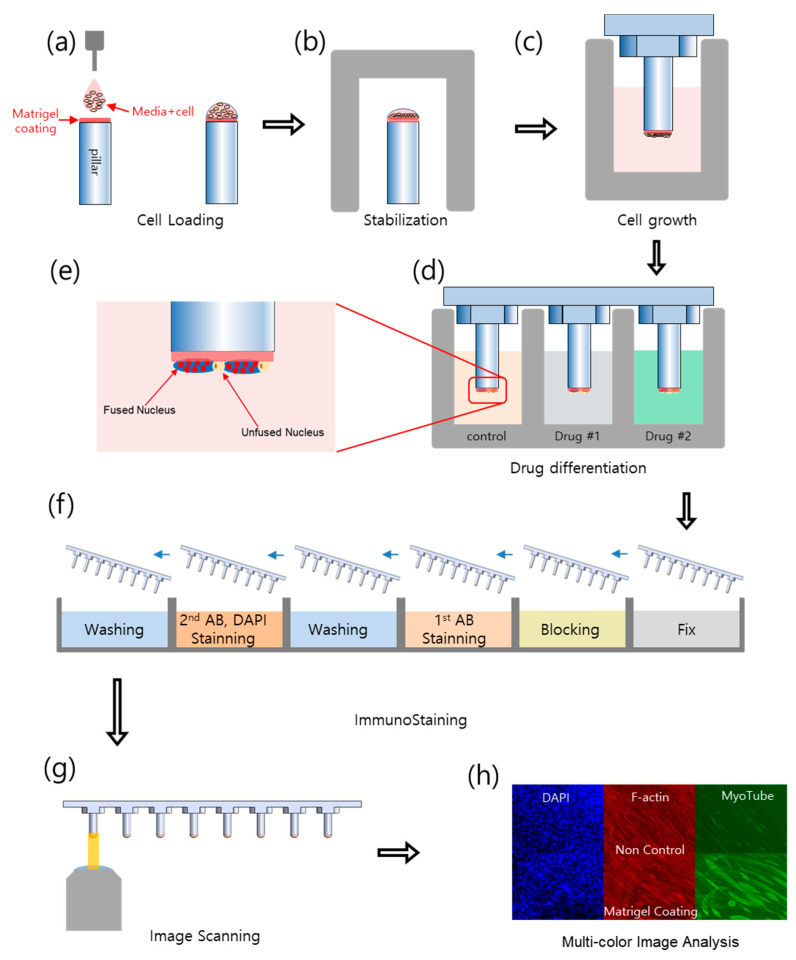
Multi-color immunostaining of C2C12 cells attached to the Matrigel-coated pillar strip. (**a**) C2C12 cells and media mixture were dispensed on a Matrigel-coated pillar strip. (**b**) Cells were allowed to attach to the pillar strip for 1.5 h. (**c**) Pillar strips (and attached cells) were dipped into wells filled with media for 2 days. (**d**) Pillar strips were dipped into the myotonic media containing six drugs for five days. (**e**) Enlarged view of pillar surface illustrating fused and unfused nucleus. (**f**) Immunostaining of the pillar strip through sequential multi-color immunostaining steps. (**g**) Multi-color image scanning of the stained C2C12 cells. (**h**) Multi-color image analysis.

**Figure 3 molecules-26-05805-f003:**
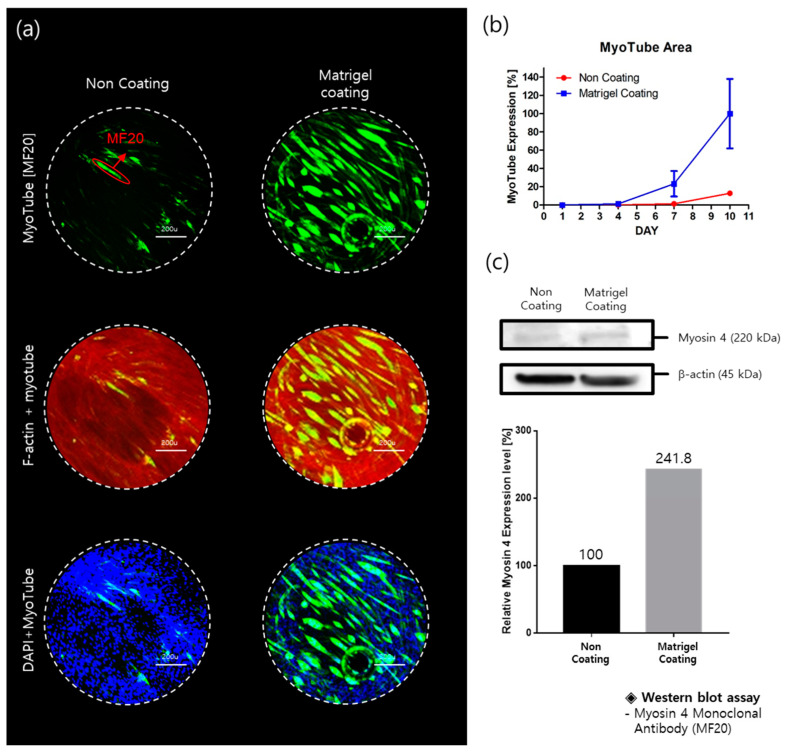
Relative expression of myotube (MF20) as assessed using a pillar strip. (**a**) A comparison of C2C12 cells cultured on uncoated and Matrigel-coated wells. (**b**) Myotube (MF20) expression extracted from the green fluorescence area of the non-Coating and Matrigel-Coating conditions. (**c**) Myotube (MF20) Western blot assay of the non-coating and Matrigel-coating conditions.

**Figure 4 molecules-26-05805-f004:**
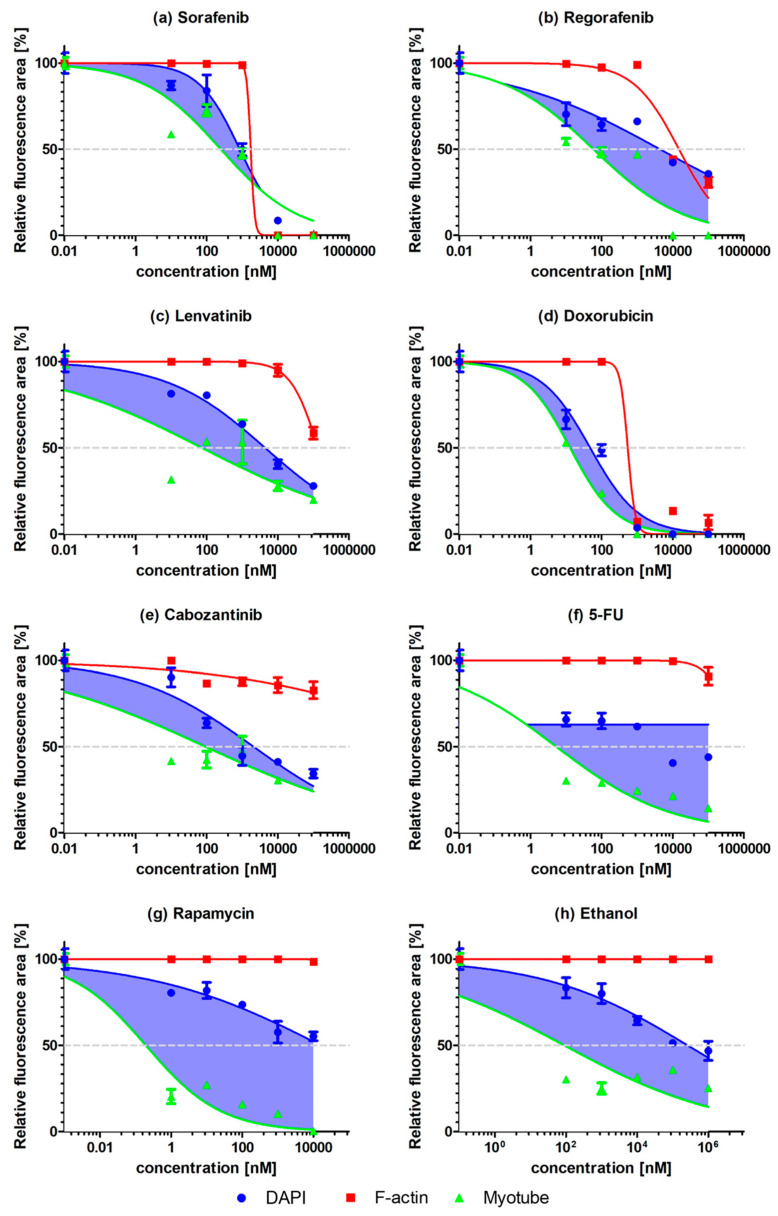
Dose-response curves of eight drugs based on relative fluorescence area of green (MF20), red (F-actin), and blue (Nuclei). (**a**) Sorafenib. (**b**) Regorafenib. (**c**) Lenvatinib. (**d**) Doxorubicin. (**e**) Cabozantinib. (**f**) 5-FU. (**g**) Rapamycin. (**h**) Ethanol.

**Figure 5 molecules-26-05805-f005:**
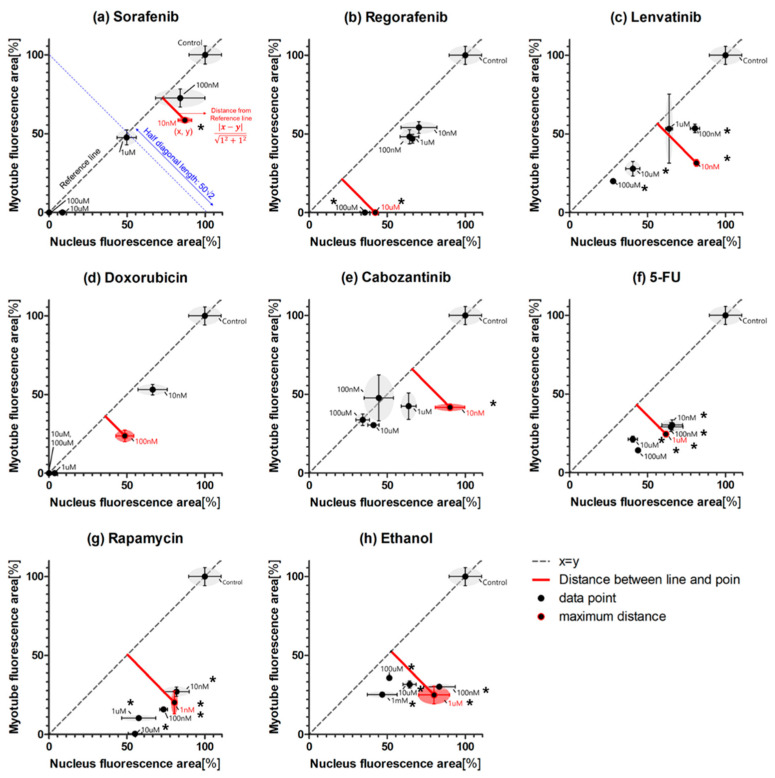
Distribution of myotube and nucleus fluorescence areas at different drug concentrations. The black dashed line (reference line) indicates the 50:50 ratio of myotubes and nuclei. The red solid line represents the maximum distance between the reference line and data points. “*” means the *p*-value between nucleus and myotube is less than 0.01. (**a**) Sorafenib. (**b**) Regorafenib. (**c**) Lenvatinib. (**d**) Doxorubicin. (**e**) Cabozantinib. (**f**) 5-FU. (**g**) Rapamycin. (**h**) Ethanol.

**Figure 6 molecules-26-05805-f006:**
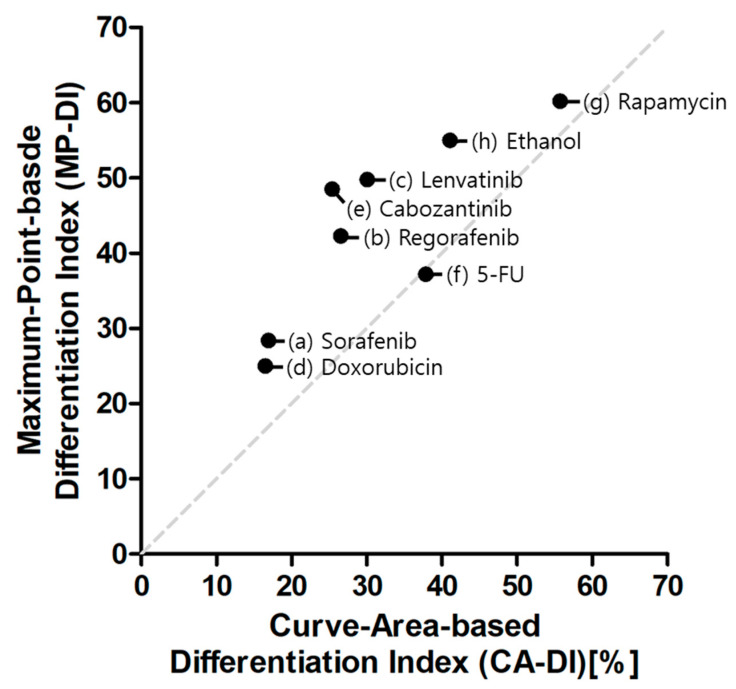
The comparison of two differentiation indexes (CA-DI and MP-DI).

**Table 1 molecules-26-05805-t001:** AUC and differentiation indices for eight drugs tested.

	CA-DI			MP-DI			
	Myotube [A.U.]	Nucleus [A.U.]	Differentiation Index [%]	Drug Concentration	Nucleus [%]	Myotube [%]	Differentiation Index [%]
1. Sorafenib [22]	387.4	466.4	16.93	10 nM	87.0	58.5	28.4
2. Regorafenib [24]	353.5	481.1	26.52	10 µM	42.3	0	42.3
3. Lenvatinib [25]	357.8	511.5	30.04	10 nM	81.4	31.6	49.8
4. Doxorubicin [23]	280	335.4	16.51	100 nM	48.6	23.7	25.0
5. Cabozantinib [26]	370.9	497.2	25.40	10 nM	90.2	41.7	48.5
6. 5-FU [27]	292.7	470.8	37.82	1 µM	61.7	24.5	37.2
7. Rapamycin [21]	244.2	551.5	55.72	10 µM	80.4	20.3	60.2
8. Ethanol [28]	315.7	535.8	41.07	100 µM	79.9	25.0	55.0

## Data Availability

Not applicable.

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
