# Peer review of "A Pillar-Based High-Throughput Myogenic Differentiation Assay to Assess Drug Safety"

_molecules, 2021, doi:10.3390/molecules26195805_

Round 1
Reviewer 1 Report
Though there are typos and awkward terms, the manuscript can be published after minor revisions that are annotated in the attached pdf file.

Author Response
#1 Reviewer’s comments
1)add the following paper which reported the use of cell calcium measurement to indicate drug toxicity of daunorubicin to muscle cells.
Electrophoresis, 2007, 28, 4723-4733.
Author’s Response) We add reference.
2) 8 pillars dipped into six wells; what about the 2 remaining wells? Layout about five conc from 10 nM to 100 uM??
Author’s Response) Side wells filled with PBS to prevent to evaporate media during cell culture. We add this sentence in 78 line. And we revised drug concentration in Figure 1.
3) Equation number?
Author’s Response) We revised equation numbers
4) In 145 line 50√2 should be illustrated using diagram to understanding how this values is obtained.
Author’s Response) We indicate half diagonal length (50√2) in Figure 5a.
5) what is the purpose of the plasma treatment?
Author’s Response) Plasma activates the polystyrene of pillar to enhance adhesion of matigel. We add this sentence in 216 line.
6) what are the pillars made of?
Author’s Response) Pillar-strips made of polystyrene. We add this sentence in 215 line.
7) In 270 line, use a diagram to illustrate what the curves are and what the gap area is?
Author’s Response) We indicate the gap area in Fig.4.
8) two equations should be illustrated with diagram.
Author’s Response) We revised Fig.4 and Fig.5a to illustrate two equations.
9) What is difference between Beta actin and F-actin?
Author’s Response) Beta actin is control for Weston plot which are used conventional. F-actin is cell skeleton in immunostaining. We used it to check cell morphology of C2C12.

Reviewer 2 Report
In this work, the authors report a high-throughput, columnar pillar strip-based assay for screening drug safety for developmental toxicity without the need for extensive washing steps and without cell loss. Meanwhile, the authors using this method, analyze eight drugs at six doses,through comparing these F-actin nucleus and myotube data, the authors proposed two differentiation indices, including Curve-Area-based Differentiation Index (CA-DI) and Maximum-Point-basde Differen-tiation Index (MP-DI).Both indices can screen of high-myogenic inbibitory drugs and the MP-DI show the senstivity for quantifying drugs which inhibited myogenic differentiation. It is an interesting topic for researchers in the related areas but the data is insufficient and the paper needs very significant improvement before acceptance for publication. My comments are summarized below:
- This paper does not compare the proportion of cell loss and the effect on the yield of this method of using pillar strip-based assays and the precious common method, hoping to add experiments for analysis, which is more conducive to the promotion of this new method.
- In Figure 4, The correlation of the dose-response curves of the eight drugs based on the Myotube relative fluorescence area is not very strong, so suggest authors to repeat the experiment and add the experimental group which can obtain a more realistic fitting data curve.On the other hand,the Figure 4 should add a data graph of the control group without any drugs to facilitate the reader's comparative analysis.
- In Figure 5, one of the eight drugs at six doses should be analyzed by conventional methods to verify the accuracy of the MPDI index, while the analysis of the other seven drugs can be included in the supporting information.
- The whole experiment requires a very long pretreatment time, I recommend to try to shorten the time and optimize the steps so as to understand whether substances will affect cell differentiation faster and better.
Author Response
#2 Reviewer’s comments
In this work, the authors report a high-throughput, columnar pillar strip-based assay for screening drug safety for developmental toxicity without the need for extensive washing steps and without cell loss. Meanwhile, the authors using this method, analyze eight drugs at six doses,through comparing these F-actin nucleus and myotube data, the authors proposed two differentiation indices, including Curve-Area-based Differentiation Index (CA-DI) and Maximum-Point-basde Differen-tiation Index (MP-DI).Both indices can screen of high-myogenic inbibitory drugs and the MP-DI show the senstivity for quantifying drugs which inhibited myogenic differentiation. It is an interesting topic for researchers in the related areas but the data is insufficient and the paper needs very significant improvement before acceptance for publication. My comments are summarized below:
1)This paper does not compare the proportion of cell loss and the effect on the yield of this method of using pillar strip-based assays and the precious common method, hoping to add experiments for analysis, which is more conducive to the promotion of this new method.
Author’s Response) Before starting experiment using pillar-strip, we coated Matrigel on the surface of the pillar to optimized C2C12 differentiation. We tried to coat wells of 96 well plate with Matrigel and conducted C2C12 differentiation experiment. However, most of the cells are clustered at the edges of the wells, and C2C12 is easily detached, especially during immunostaining at multiple aspiration and filling steps. We think this detachment depended on user skill and equipment. And most C2C12 immunostaining was conducted on cover glass. We could not afford to conduct many tests with cover glass. And editor of Molcules journal give one week to revision. Unfortunately, we can not get additional data. We add “Before starting experiment using pillar-strip, we tried to coat wells of 96 well plate with Matrigel and conducted C2C12 differentiation experiment. However, most of the cells are clustered at the edges of the wells, and C2C12 is easily detached, especially during immunostaining at multiple aspiration and filling steps. This detachment may vary de-pending on the user’s skill and equipment.” in 105 line.
2) In Figure 4, The correlation of the dose-response curves of the eight drugs based on the Myotube relative fluorescence area is not very strong, so suggest authors to repeat the experiment and add the experimental group which can obtain a more realistic fitting data curve. On the other hand, the Figure 4 should add a data graph of the control group without any drugs to facilitate the reader's comparative analysis.
Author’s Response) As Reviewer’s comments, Myotube relative fluorescence are did not strong. However, under different concentration condition, the trend could be observed. We focus on illustrating the gap between Myotube and Nucleus fluorescence area. We revised Figure 4. The lowest drug dose in the dose response curve refers to the control (no drug status) because the x-axis of the dose response curve uses log(drug concentration) and log(0) does not exist.
3) In Figure 5, one of the eight drugs at six doses should be analyzed by conventional methods to verify the accuracy of the MPDI index, while the analysis of the other seven drugs can be included in the supporting information.
Author’s Response) Unfortunately, we did not have data. we tried to coat wells of 96 well plate with Matrigel and conducted C2C12 differentiation experiment. However, most of the cells are clustered at the edges of the wells, and C2C12 is easily detached, especially during immunostaining at multiple aspiration and filling steps. So, we fail to experiment data using conventional 96 well plate. Thus, we revised references about eight drugs in Table 1.
4) The whole experiment requires a very long pretreatment time, I recommend to try to shorten the time and optimize the steps so as to understand whether substances will affect cell differentiation faster and better.
Author’s Response) In our C2C12 cell line, Cell differentiation slowly occur even though Matrigel-coating condition. In conventional 96 well plate for cell culture, C2C12 did not differentiation or very slowly differentiation. So, we optimized pillar coating and culture times with our C2C12. For reduce assay time, increasing seeding density upto 4000cells/pillar may reduce C2C12 confluency (When the pillar surface is more than 80% filled with C2C12) time. We add this sentence in 236 line.
